# HYBRID CLOUD-EDGE NETWORKS FOR EFFICIENT INFERENCE

## ABSTRACT

Although deep neural networks (DNNs) achieve state-of-the-art accuracy on large-scale and fine-grained prediction tasks, they are high capacity models and often cannot be deployed on edge devices. As such, two distinct paradigms have emerged in parallel: 1) edge device inference for low-level tasks, 2) cloud-based inference for large-scale tasks. We propose a novel hybrid option, which marries these extremes and seeks to bring the latency and computational cost benefits of edge device inference to tasks currently deployed in the cloud. Our proposed method is an end-to-end approach, and involves architecting and training two networks in tandem. The first network is a low-capacity network that can be deployed on an edge device, whereas the second is a high-capacity network deployed in the cloud. When the edge device encounters challenging inputs, these inputs are transmitted and processed on the cloud. Empirically, on the ImageNet classification dataset, our proposed method leads to substantial decrease in the number of floating point operations (FLOPs) used compared to a well-designed high-capacity network, while suffering no excess classification loss. A novel aspect of our method is that, by allowing abstentions on a small fraction of examples ($< 20\%$), we can increase accuracy without increasing the edge device memory and FLOPs substantially (up to 7% higher accuracy and 3X fewer FLOPs on ImageNet with $80\%$ coverage), relative to MobileNetV3 architectures.

## 1 INTRODUCTION

Deep Neural Networks (DNNs) achieve state-of-the-art (SOTA) performance on challenging tasks such as image recognition (Tan & Le, 2019; Howard et al., 2019), language modelling (Devlin et al., 2018), and machine translation (Wu et al., 2016b). High accuracy on such tasks often comes at a high memory and compute cost, making DNN deployment on low resource edge hardware like microcontroller units (MCUs) very challenging (Banbury et al., 2021; Fedorov et al., 2019; Lin et al., 2020; Fedorov et al., 2020; Gural & Murmann, 2019).

**Edge-Device and Cloud ML** As such, two paradigms currently co-exist. Edge-device inference utilizes lightweight architectures and focuses on low-level tasks such as smart messaging and face recognition (Google LLC, 2021). In parallel, more complex and nuanced tasks are deployed in the cloud, which is a term we use to refer to over-provisioned hardware platforms like a GPU server (Alemi, 2016). The fundamental drawback of cloud ML, however, is the increased latency and energy consumption arising from communication, which can be prohibitive for many applications. Indeed, the meager cost, size, and power requirements of MCUs make them the platform of choice for a large number of applications, so that MCU shipments outnumber GPU shipments by roughly 50 to 1 (Fedorov et al., 2019; Lin et al., 2020).

While methods like pruning (Molchanov et al., 2017; Han et al., 2015), quantization (Jacob et al., 2018), knowledge distillation (Hinton et al., 2015), and adaptive computation (Bejnordi et al., 2019) could be leveraged to reduce the size of cloud-based models, these strategies fundamentally limit the resulting achievable accuracy due to the reduced model capacities (Fedorov et al., 2019; Lin et al., 2020). For example, SOTA accuracy on ImageNet is 84.3% (Tan & Le, 2019) with 37B FLOPs; whereas the best deployable model on an STM32F746 MCU under 5 frames per second constraint achieves 51.1% accuracy with 12.8M FLOPs (Lin et al., 2020) .

**Hybrid Edge-Cloud Inference.** Motivated by these emerging trends, we propose a best-of-both hybrid solution, which allows for deploying cloud-based AI tasks on edge devices like MCUs, while lowering the average total latency (the sum of communication and computational latency).

Our end-to-end hybrid approach involves architecting and training three distinct networks in concert, individually named 'base', 'global', and 'routing' (Fig. 1). The base model is compact and designed for devices with low-resource hardware constraints like latency and memory usage. In contrast, the global model has a large capacity and is deployed in the resource-rich cloud environment. Finally, the routing model, which is very compact, is used to decide whether a query should be communicated to the global model, or handled entirely by the base model, thus enabling the two to work in tandem to maximise performance while controlling usage of computational or communication resources (see Fig. 3). Ideally, the global and base models are fine-tuned to be most accurate on the queries routed specifically to them, while the routing model in turn only issues queries to the global model when they are too hard to be processed by the base model.

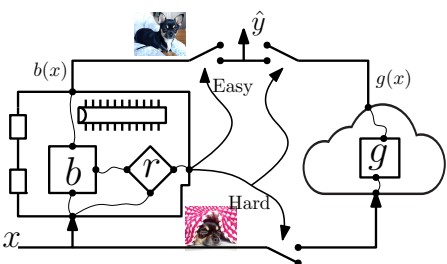

Figure 1: HYBRID MODEL. Cheap base ($b$) and routing models ($r$) run on, e.g., a microcontroller; Expensive global model ($g$) runs on, e.g., a cloud server. $r$ uses $x$ and features of $b$ to decide if $g$ is evaluated or not.

**Technical Contributions.**

*Training Methodology.* Learning an efficient hybrid model requires us to solve the challenging problem of discovering regions of 'easy' queries, where classification can reliably be performed by a simple model. In addition, training is challenging since a choice of base and global model affects the optimal routing, which cyclically affects the former. We propose an alternating-optimisation scheme that can be modularly executed, and design an efficient proxy supervision for the router, which together allow for simplified training. The result is a flexible scheme that can be used either to train all three models or only a subset. The routing model is further designed with a flexible assignment criterion that allows efficiently trading-off between a range of accuracy and resource usages, thus yielding a variety of operating points with a single training round.

*Neural Architecture Search (NAS).* Hybrid design also raises a novel architectural issue - while present DNN architectures are aligned towards a single model that performs end-to-end inference, a hybrid scheme may require a coupled design for the base and global models to best exploit the available resources. To discover efficient joint designs, we propose a NAS method that utilises an efficient proxy score to quickly determine fitness of a pair of base and global architectures, and performs an evolutionary search to optimise the accuracy at any given combined resource usage. Our approach is flexible and can also be used to, e.g., adapt a base architecture to a given global model.

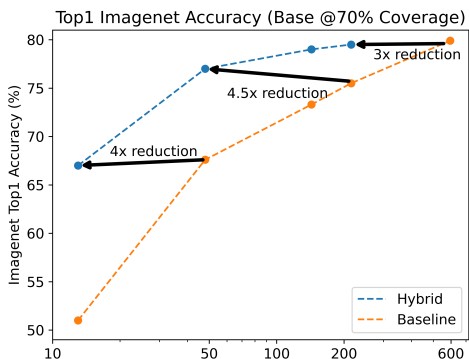

Figure 2: Base FLOP gains in the hybrid system at different levels of accuracy w.r.t. a stand-alone model, collated from Tables 1 and 3 (see Appendix §B.5 for details); 3 in 10 examples routed to a cloud model, a 70% reduction in communication latency w.r.t. purely cloud-based service.

*Empirical Validation.* Proposed above is a novel *end-to-end* methodology to design and train resource-efficient hybrid architectures and models that can perform at SOTA accuracy whilst satisfying hardware constraints. Extensive experimentation on the ImageNet dataset shows that the resulting scheme *pareto* dominates methods that learn a single efficient architecture, demonstrating $2-3.5\%$ accuracy gains at any FLOP count when compared to prior efficient architecture designs such as MobileNetV3 and OFA ( Fig. 3, 4). Further, in settings where the global device has much higher compute capacity (hence negligible inference cost), we show that whilst processing 70% of queries at the base, our design can match the accuracy of these designs with up to $4.5\times$ improvement in base FLOPs (Fig. 2).

RELATED WORK

**Efficient Architectures.** Previous works have designed low complexity DNNs for mobile applications (Iandola et al., 2016; Gholami et al., 2018; Sandler et al., 2018; Howard et al., 2019) using low-rank decomposition, separable convolutions, and hand-crafted feature blocks. Tan & Le (2019) study the effect of width, depth, channels, and input resolution on DNN memory and FLOP costs. In parallel, neural architectures searches under constraints such as FLOP (Liu et al., 2017; Zoph & Le, 2016; Dong & Yang, 2019; Elsken et al., 2019), latency (Cai et al., 2020), memory (Fedorov et al.,

2019), etc have been carried out. These efforts are complementary to our hybrid scheme since we can leverage these improved architectures as base models and achieve similar gains in performance.

**Low Compute Transformations.** Researchers have explored methods to obtain low-capacity models from SOTA DNNs, including compression and pruning (Han et al., 2016), quantization (Wu et al., 2016a), hashing (Chen et al., 2015), and knowledge distillation(Hinton et al., 2015). Since these transformations are orthogonal to our proposal and can be leveraged post-hoc, we do not pursue these techniques in order to simplify our exposition.

**Adaptive Neural Networks.** Han et al. (2021) present a comprehensive survey on designing dynamic neural networks that budget more computational resources for harder examples. These include (a) cascade-based early exit networks (Park et al., 2015; Bolukbasi et al., 2017; Nan & Saligrama, 2017; Wang et al., 2018), where the constituent networks are independently designed and do not share intermediate features; (b) early exit networks (Teerapittayanon et al., 2017; Dai et al., 2020; Li et al., 2019) where classifiers are introduced at intermediate layers; and (c) multi-scale networks with early exits (Huang et al., 2018; Yang et al., 2020), which are allowed to operate at different input resolution, width or depth. Kang et al. (2017) splits network execution between device and cloud, resulting in higher communication as the features require more storage than input. Further, a high-performing model can neither be stored nor executed on a constrained MCU due to low RAM and Flash.

In the context of our problem, much of the focus in these works is on scaling-up capacity without a proportional increase in inference time. Although our proposed hybrid approach bears resemblance to these works, we are focused on the opposite scenario, namely, how to overwhelmingly reduce resource usage (FLOPs and communication latency) to allow for deployment on edge devices without degrading accuracy achievable by a large model. As such, our perspective necessitates posing an end-to-end system-wide hybrid objective and requires systematic integration and optimization of all of the degrees of freedom (architectures, routing & coverage, base and global networks). In contrast, prior works optimize these aspects in a decoupled and isolated manner. As a case in point, works focusing on architectures and early exit networks utilize simple entropy thresholding for routing. As we will show in our results, carefully designed routing schemes, which are jointly optimized along with base and global models, can result in substantial gains over entropy thresholding. Li et al. (2021) model a hybrid system with a similar design but missing crucial details, namely, (a) no coverage penalty in the train loss, (b) router is entangled with the base and global network, while we decouple it using the routing oracle, and (c) no evaluations on Imagenet (see Appendix §B.7 for details).

**Learning with Abstention.** Many researchers (Liu et al., 2019; Gangrade et al., 2021; Geifman & El-Yaniv, 2019) have studied the problem of learning with a reject option, where a model can abstain prediction on some examples with the goal of minimizing the number of errors and abstentions. Although we get an abstaining classifier from the hybrid model by simply ignoring the global model, our main objective is to improve the performance of the hybrid system that includes the global model.

## 2  METHOD

Let $\mathcal{X}$ be a feature space and $\mathcal{Y}$ a set of labels. A hybrid design is composed of three models:

- A *base model* $b : \mathcal{X} \to \mathcal{Y}$, that can be deployed on an edge device.
- A *global model* $g : \mathcal{X} \to \mathcal{Y}$, that is deployed in the cloud and typically has high accuracy.
- A *routing model* $r : \mathcal{X} \to \{0, 1\}$, that is a very low resource model deployed alongside the base model, and routes hard queries to the global model.

We will treat these models as soft classifiers, outputting $|\mathcal{Y}|$-dimensional scores $\{b_y\}$ and $\{g_y\}$, and two scores $r_0(x)$ and $r_1(x)$ for the routing model. In this paper, $r$ is realized by a 2-layer DNN with input $b_y(x)$. The default hard output for the base is the top entry $b(x) = \arg\max_y b_y(x)$, and similarly for $g$. By default $r$ assigns $x$ to the global model if $r_1(x) > r_0(x)$, i.e., $r(x) = \mathbb{1}\{r_1(x) > r_0(x)\}$, but this can be relaxed to $r(x; t) := \mathbb{1}\{r_1(x) > t + r_0(x)\}$, where the hyper-parameter $t$ allows a routing model to trade-off accuracy and resource usage in order to avoid separately training for each desired level. The decision produced by the system for an instance $x \in \mathcal{X}$ is

$$\hat{y}(x) := (1 - r(x))b(x) + r(x)g(x). \qquad (1)$$

The *coverage* of the hybrid system is the fraction of instances that are processed by the base only, i.e.

$$\mathcal{C}(r, b, g) := \mathbb{P}(r(X) = 0),$$

where $\mathbb{P}$ denotes the joint law over $(X, Y)$. The *hybrid accuracy* is

$$\mathcal{A}(r, b, g) = \mathbb{P}(\hat{y}(X) = Y) = \mathbb{P}(r(X) = 0, b(X) = Y) + \mathbb{P}(r(X) = 1, g(X) = Y).$$

**Architectures and Costs.** The resources required to evaluate a DNN are mainly a function of its architecture - the number and arrangement of its layers and weights. We use $\alpha$ to denote a generic architecture, and say that a model $f \in \alpha$ if it is realizable by this architecture. To quantify the resource consumption, let $\mathcal{R}(\alpha)$ denote the cost per inference for a model with architecture $\alpha$. In our design, the base is always executed, with the output fed into the routing model. Therefore, the hybrid FLOP count of a hybrid model $(r, b, g)$, such that $b \in \alpha_b$ and $g \in \alpha_g$, is

$$\mathcal{R}(r, b, g) := \mathcal{R}_r + \mathcal{R}(\alpha_b) + (1 - \mathcal{C}(r, b, g))\mathcal{R}(\alpha_g) \tag{2}$$

where $\mathcal{R}_r$ is a fixed, small quantity required to execute $r$. We can model many resources, including the FLOPs required to execute the model, and the latency on edge devices like MCUs (Banbury et al., 2021). Additionally, for settings where the global model is on the cloud with a compute-rich environment, the resource costs are dominated by communication latency, modeled as $\mathcal{R}(\alpha_g) = \tau, \mathcal{R}(\alpha_b) = 0$ (where $\tau$ is the mean communication delay). We use coverage as a proxy metric for communication latency as it measures the data split between the base and global. In the following, we will focus on FLOP and coverage metrics, although we investigate inference latency in §3.4..

**Overall Formulation** Let $\mathscr{A}_b$ and $\mathscr{A}_g$ be sets of base and global architectures, which may incorporate implementation restrictions, and $\varrho$ a target resource usage level. Our objective is

$$\max_{\alpha_b \in \mathscr{A}_b, \alpha_g \in \mathscr{A}_g} \max_{r, b \in \alpha_b, g \in \alpha_g} \mathcal{A}(r, b, g) \quad \text{s.t.} \quad \mathcal{R}(r, b, g) \leq \varrho. \tag{3}$$

The outer maximisation over $(\alpha_b, \alpha_g)$ in (3) amounts to an architecture search, while the maximisation over $(r, b, g)$ in a fixed architecture corresponds to learning a hybrid model. The following sections describe our method for solving (3). Briefly, we propose to decouple the inner and outer optimization problems in (3) for the sake of efficiency - hybrid models are trained by an empirical risk minimisation (ERM) strategy, whilst the architecture search is carried out using fast proxies for the accuracy attainable by a given pair of architectures without directly training hybrid models.

## 2.1 LEARNING HYBRID MODELS

This section focuses on training hybrid models for fixed architectures $\alpha_b, \alpha_g$, i.e., the inner problem

$$\max_{r, b \in \alpha_b, g \in \alpha_g} \mathcal{A}(r, b, g) \quad \text{s.t.} \quad \mathcal{R}(r, b, g) \leq \varrho. \tag{4}$$

Since architectures are fixed in (4), the FLOP constraint amounts to a constraint on the hybrid coverage. As is standard, we will approach (4) via an ERM over a Lagrangian of relaxed losses. However, a number of design considerations and issues need to be addressed before such an approach is viable, as discussed below. The overall scheme is summarised in Algorithm 3 in §A

**Alternating optimisation.** Problem (4) has a cyclical non-convexity. A given $r$ affects the optimal $b$ and $g$ (since these must adapt to the regions assigned by $r$), and vice-versa. We approach this issue by alternating optimisation. First, we train global and base models according to standard methods. Then, we learn a routing network $r$ under a coverage penalty. The resulting $r$ feeds back into the loss functions of $b$ and $g$, and these models get retrained. This cycle may be repeated many times.

*Modularity of training*. Resulting scheme allows training $r$ with a fixed $(b, g)$, as it helps learn a cheap routing model with pre-trained cloud and mobile models. Similarly, by dropping the optimisation over $g$, we can hybridise too expensive to re-train global models. Additionally, one can initially train the global and freeze it after a few cycles to save compute. Finally, we can learn each component to different degrees, e.g., we may take many more gradient steps on $g$ than $r$ or $b$ in any training cycle.

**Learning Routers via Proxy Supervision.** Given a fixed pair of base and global model $(b, g)$, the problem (4) reduces to the following, where $C_\varrho$ is the coverage needed to ensure $\mathcal{R} \leq \varrho$.

$$\max_r \mathbb{E}[(1 - r(X))\mathbb{1}\{b(X) = Y\} + r(X)\mathbb{1}\{g(X) = Y\}] \quad \text{s.t.} \quad \mathbb{E}[r(X)] \leq C_\varrho. \tag{5}$$

While a naïve approach is to relax $r$ and pursue ERM, we instead reformulate the problem. Observe that (5) demands that $r(X) = 0$ if $b(X) = Y, g(X) \neq Y$, and that $r(X) = 1$ if $b(X) \neq Y, g(X) = Y$. Further, while case $b(X) = g(X) = Y$ is not differentiated, the coverage constraint promotes $r(X) = 0$ for such points. Thus, the program can be viewed as a supervised learning problem of fitting the *routing oracle*, i.e.

$$o(x; b, g) = \mathbb{1}\{b(x) \neq g(x) = y\}. \tag{6}$$

Indeed, $o$ is the ideal routing without the FLOP constraint. It can be evaluated on training data - for any given $(b, g)$ and training dataset $\mathcal{D} = \{(x^i, y^i)\}$, we produce the oracle dataset $\mathcal{D}_{o;(b,g)} := \{(x^i, o(x^i; b, g))\}$. We use this dataset as supervision for the routing model $r$, which allows us to utilise the entire gamut of tools of machine learning that are essential for practically learning good binary functions, thus gaining over approaches that directly try to relax the objective of (5).

Note that the oracle $o$ does not respect the FLOP constraint. We can satisfy such a constraint by randomly assigning some points from $g$ to $b$ while incurring an error. The oracle is indifferent to such an arrangement. From a learnability perspective, we would like the points flipped to the base to promote regularity in the dataset. Although, such a goal is ill-specified (and unlikely to be captured well by simple rules such as ordering points by a soft loss of $g$). Instead, we handle this issue indirectly by imposing a coverage penalty whilst training the routing model and leave it to the optimisation to discover the appropriate regularity by minimising error w.r.t. $o$ under this penalty.

**Focusing competency and loss functions**. To improve accuracy whilst controlling coverage, we focus the capacity of each of the models on the regions relevant to it - so, $b$ is biased towards being more accurate on the region $r^{-1}(\{0\})$, and similarly $g$ on $r^{-1}(\{1\})$. Similarly, for the routing network $r$, it is more important to match $o(x)$ on the regions where it is 1, since these regions are not captured accurately by the base and thus need the global capacity. We realise this inductive bias by introducing model-dependent weights in each loss function to emphasise the appropriate regions. The *Routing Loss* consists of two terms, traded off by a hyperparameter $\lambda_r$ - the first penalises deviation of coverage from a given target (cov), and the second to promote alignment with $o$ and is biased by the weight $W_r(x) = 1 + 2o(x)$ to preferentially fit $o^{-1}(\{1\})$. The symbol $\ell$ denotes a surrogate loss (such as cross entropy), while $(\cdot)_+$ is the ReLU function, i.e., $(z)_+ = \max(z, 0)$. All sums below are over a dataset $\mathcal{D} = \{(x^i, y^i)\}$ of size $N$. Empirically, we find that $\lambda_r = 1$ produces effective results.

$$\mathcal{L}_{\text{routing}}(r; o) := \lambda_r \left( \text{cov} - \left( 1 - \frac{1}{N} \sum_x (r_1(x) - r_0(x))_+ \right) \right)_+ + \sum_x W_r(x) \ell(o(x), r(x)). \quad (7)$$

The *Base Loss* and the *Global Loss* are each a weighted variant of the standard classification loss, which are biased by the appropriate weights to emphasise the regions assigned to either model by the routing network - $W_b(x) = 2 - r(x)$ and $W_g(x) = 1 + r(x)$.

$$\mathcal{L}_{\text{base}}(b; r, g) = \sum W_b(x) \ell(y, b(x)), \quad \text{and} \quad \mathcal{L}_{\text{global}}(b; r, g) = \sum W_g(x) \ell(y, g(x)).$$

## 2.2 EVOLUTIONARY ARCHITECTURE SEARCH

This section describes the joint architecture search implicit in (3). We use an evolutionary search(Elsken et al., 2019; Liu et al., 2021) due to its simplicity and effectiveness. It requires a fast way to evaluate the fitness of a base-global pair $(\alpha_b, \alpha_g)$, i.e. the value of the program (4) for a given pair of architectures. While this can be obtained by carrying out hybrid training as previously described it is impractical due to the time cost. The following describes a cheaper proxy for the same.

To quickly approximate the result of training over $r$, we use the agreement oracle routing $\hat{o}$. This assigns all inputs for which the base and global agree to the base, and the remainder to the global model. We choose the agreement oracle as opposed to the oracle in (6), as it is more realistic in that it does not assume knowledge of the true classification label, while still being easy to compute.

What remains is the optimisation over $b$ and $g$. We adopt a different solution for these, rooted in the space of architectures itself. Recently, Cai et al. (2020) showed that it is possible to design spaces of architectures such that each architecture $\alpha$ is associated with a canonical set of parameters $\theta_\alpha$ that are near-optimal in an accuracy objective, in the sense that a slight fine-tuning of these models yields a good solution. This is realised via the use of a 'super-net,' the components of which can be individually changed, leading to a combinatorially large set of architectures. Importantly, training a super-net with such a property has comparable costs to training a standard network, and this cost is further amortized over a large number of resulting architectures. We use OFA space of Cai et al. (2020) as architectural search space, and incorporate the oracle proxy to avoid training $r$, yielding the score $\mathcal{A}(\hat{o}, \theta_{\alpha_b}, \theta_{\alpha_g})$ for the fitness of $(\alpha_b, \alpha_g)$. The resulting architecture search scheme is summarised in Alg. 4 in §A.

## 3 EXPERIMENTS

In this section, we will first evaluate hybrid models with off-the-shelf base and global architectures. Next, we will perform evolutionary search to find architectures under various resource constraints and evaluate hybrid models with these newly found architectures. Finally, we infer that accuracy of the hybrid model consistently increases with increasing differences between base and global FLOPs.

For a proof-of-concept, we limited the cloud model to 600M FLOPS, which has a SOTA accuracy of $\approx 80\%$ (Cai et al., 2020). While there are other models such as EfficientNet achieving higher accuracy (84.3%), these require substantially more FLOPs (37B). Although our hybrid training could leverage such models, our limited computing resources made this infeasible.

**Highlights.** We list a few salient observations from our empirical results.

- *Pareto Dominance and Latency Reduction.* Hybrid models consistently outperform high-capacity models at lower Hybrid FLOPs (Fig. 3) and at a latency reduction of 70% (3 in 10 examples pass to the cloud) our FLOP gains are substantial ( Fig. 2). Similarly, we improve accuracy at the same FLOP level both in terms of base FLOPs and Hybrid FLOPs. For instance, 80% is SOTA accuracy for a stand-alone 600M model. We achieve 79% accuracy in a Hybrid scheme with a base model of 143M (see Table. 1) and 78.5% with 350M hybrid FLOPs.
- *Rapid Customization.* Proposed approach allows for optimizing accuracy level to match any intermediate FLOP count with little training, saving computation for training intermediate models.
- *SOTA performance on Resource Constrained Base.* Hybrid scheme allows the base model to be deployable on a low-resource hardware. With 12M base FLOPs and only 3 in 10 examples passed to a larger model we gain about 16% improvement in accuracy (see 3).
- *Evolutionary Search yields better Hybrid Models.* Given any single model, we can obtain a better hybrid model using evolutionary architecture search with similar FLOP count.
- *Routing outperforms Entropy Metric.* Our routing method exploits base and global models characteristics and dominates entropy-thresholding used in many adaptive neural networks.

**Experimental Setup.** For simple exposition, we focus on the classification task on the Imagenet (Russakovsky et al., 2015) dataset, consisting of 1.28M train and 50K validation images. We follow standard data augmentation (mirroring, resize and crop to shape $224 \times 224$) for training and single crop for testing. Similar to previous works, we report results on the validation set. We borrow the pre-trained baselines from their public implementations as described in the appendix sec. B.2. For evolutionary architecture search, we utilize the supernet from the OFA search space (Cai et al., 2020). We describe our hyper-parameter settings in the appendix sec. B.1. Depending on the computational budget, one can create hybrid models in three ways (a) 'Hybrid-(r)' - only training routing while using pre-trained base and global, (b) 'Hybrid-(rb)' - training routing and base while using pre-trained global, and (c) 'Hybrid-(rbg)' - training all three components. We trained the hybrid model to achieve a similar coverage level as the oracle. Post training, we vary the coverage level by adjusting the threshold hyper-parameter in the routing to generate the performance at different hybrid FLOPs.

### 3.1 HYBRID MODELS USING OFF-THE-SHELF CLASSIFIERS

#### 3.1.1 NO RESOURCE CONSTRAINTS

In this setting, we assume no constraints on the model deployment. We pick up an architecture family and create a hybrid model using the smallest and largest architecture. For convenience, we perform this experiment for two known families, namely MobileNetV3(Howard et al., 2019) and OFA(Cai et al., 2020). From MobileNetV3, we pick the smallest model (48M FLOPs, 67.6% accuracy) as base and largest model (215M FLOPs, 75.7% accuracy) as global to create Hybrid-MobileNetV3 model. Similarly, from OFA, we pick the smallest model (67M FLOPs, 70.4% accuracy) as base and largest model (595M FLOPs, 80% accuracy) as global to create Hybrid-OFA model.

Figure 3 (a) and (b) plot the FLOPs vs top1 accuracy, and compare the hybrid models with best-known baseline models in the architecture space. We show hybrid FLOPs for the hybrid models (see Eq. 2). These experiments provide evidence for the following properties of hybrid models:

- *Hybrid models dominate any standalone model between base and global model.* Hybrid models outperform off-the-shelf classifiers at every intermediate FLOP count. For ex., a pre-trained model in MobileNetV3 with 155M FLOP achieves 73.3% accuracy while our hybrid model 'Hybrid-(rbg)'

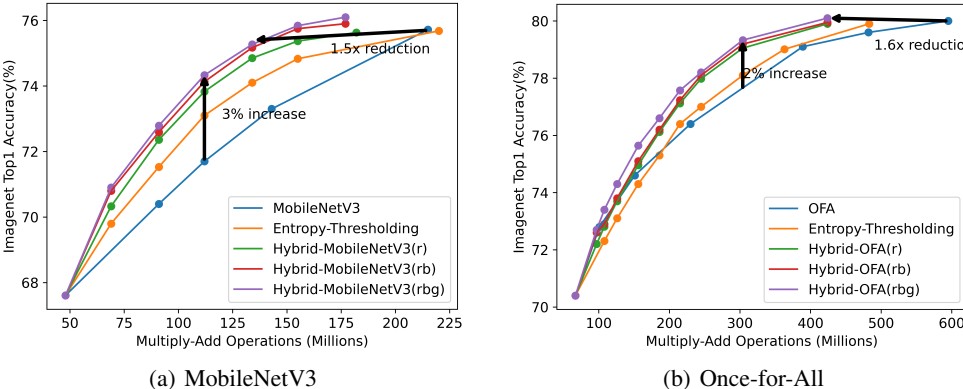

(a) MobileNetV3          (b) Once-for-All

Figure 3: Plot for FLOPs vs accuracy under no resource constraints. Each intermediate point for the baseline requires expensive training and fine-tuning. In addition, OFA requires an evolutionary search to find the model. In contrast, the proposed scheme creates hybrid models using two extreme points and achieves better performance than the best fine-tuned models in this region.

Table 1: Results for hybrid models with base at various coverage levels. OFA model achieving $\approx 80\%$ Top1 accuracy is used as global model. Base model belongs to MobileNetV3 space.

| Base MACs | Base Accuracy (%) | Coverage=90% Accuracy (%) | | Coverage=80% Accuracy (%) | | Coverage=70% Accuracy (%) | |
|---|---|---|---|---|---|---|---|
| | | Base | Hybrid | Base | Hybrid | Base | Hybrid |
| 48M | 67.61 | 73.3 | 71.59 | 78.56 | 74.61 | 83.41 | 76.77 |
| 143M | 73.3 | 79.01 | 75.94 | 83.88 | 77.81 | 88.39 | 79.01 |
| 215M | 75.72 | 81.33 | 77.61 | 86.07 | 79.01 | 90.11 | 79.59 |

achieves 75.5% accuracy. Similarly, a model in OFA with 230M FLOP achieves 76.4% accuracy while the hybrid model achieves 77.6% accuracy.

- *Hybrid achieves SOTA w.r.t a global model at $\approx 20\%$ lower FLOP count.* Global model in MobileNetV3 achieves 75.7% accuracy at 215M FLOPs, while hybrid model achieves same accuracy at 177M FLOPs. Similarly, global model in OFA achieves 79.9% accuracy at 595M FLOPs, while hybrid model achieves same accuracy at 483M FLOPs.

- *Training a Hybrid model for intermediate FLOPs is inexpensive.* To achieve a single model at any FLOPs, we find an architecture with the FLOP constraint and train it to achieve non-trivial performance. Hybrid model with smallest and largest model allows us to trade-off FLOPs for accuracy and save compute for training models for an intermediate FLOP constraint.

- *Outperform the entropy thresholding baseline used in dynamic neural networks.*

- *End-to-end training of all components lead to increasing gains.* Hybrid models improve in performance as additional components are trained in the alternative minimization (Algorithm 3), i.e. Hybrid-rbg is the best performing model followed by Hybrid-rb and Hybrid-r.

### 3.1.2 Resource Constrained Base

In this setting, our base model operates at a fixed computational budget on an edge device and one cannot deploy the global model on this device. For simplicity, we assume the latency between the base and global models to be negligible. We create hybrid models using base models from the MobileNetV3 family. Since in this setting, the goal is to save compute / battery on the device and achieve near SOTA performance, we use a high performing OFA model as the global model and operate the base model at a fixed coverage level. We report two metrics: (a) base accuracy achieved by the base by predicting only on the coverage portion of the routing, (b) hybrid accuracy - accuracy achieved by the hybrid model, where the examples abstained by the base are sent to the global model. Table 1 shows the base and hybrid accuracy at three coverage levels, 90%, 80% and 70%. Hybrid models operating at a fixed coverage level provide the following benefits:

- *Hybrid models achieve near SoTA accuracy with $\approx 3x$ less FLOPs.* Using base with 48M FLOPs and 67.61% accuracy, the hybrid model achieves 71.59% accuracy at 90% coverage, improving to 76.77% accuracy at 70% coverage. To achieve 76.77% accuracy with a single model would require > 400M FLOPs, which is too large to be deployed on an MCU. Similarly, using a base

Table 2: Results for the evolutionary search for hybrid architectures at different FLOP constraints

| | Hybrid NAS - 150M | | Hybrid NAS - 250M | | Hybrid NAS - 350M | |
|---|---|---|---|---|---|---|
| | Accuracy (%) | MACs | Accuracy (%) | MACs | Accuracy (%) | MACs |
| Base Model | 68.65 | 68M | 71.68 | 145M | 74.33 | 225M |
| Global Model | 75.98 | 263M | 78.4 | 466M | 78.75 | 501M |
| OFA Search@Flops | 73.71 | 150M | 74.77 | 250M | 74.93 | 400M |
| Entropy Thresholding | 73.83 | 150M | 76.05 | 252M | 77.2 | 350M |
| Hybrid (r) | 74.51 | 150M | 76.63 | 252M | 77.91 | 350M |
| Hybrid (rb) | 74.72 | 150M | 76.91 | 252M | 78.07 | 350M |

with 215M FLOPs and 75.7% accuracy, the hybrid model achieves 79.59% accuracy with 70% coverage . Global with ≈ 600M FLOPs has 80% accuracy.

- *Abstaining base model achieves significantly better performance than the base at full coverage.* Hybrid scheme allows the system to operate without a global model. In this case. the result is an abstaining classifier operating on the device, i.e. it rejects few input examples and provides predictions on the rest. For ex., a base model with 48M FLOP achieves an accuracy of 83.41% when it only covers 70% examples. Similarly, a base model of 215M achieves 90.11% accuracy with 70% coverage. In both cases, there is a gain of at least 15 points in the accuracy.

### 3.2 HYBRID MODELS WITH EVOLUTIONARY ARCHITECTURE SEARCH

So far we have been generating hybrid models using off-the-shelf classifiers that are not tuned to maximize hybrid performance. In this experiment, we search for base and global pairs using evolutionary search in the OFA space. We constrain the search to operate at fixed hybrid FLOPs. After finding base and global pairs from the evolutionary search, we create hybrid models with the newly found architectures. We perform this experiment for three hybrid FLOP constraints: 150M, 250M, and 350M. We draw baseline architecture samples from the OFA space using their optimized architecture search. For fair comparison, we do not fine tune models found by the architecture searches for both OFA and hybrid models. Figure 4 plots the operating curves for the hybrid models found using different FLOP constraints. Table 2 shows the hybrid model performance at the constraint points used in the search. Evolutionary search based hybrid models provide the following benefits

- *Hybrid Models with evolutionary search yields higher accuracy at any target FLOP.* As illustrated in Table 2, evolutionary search finds hybrid models that outperform the models found from the optimized OFA search. For ex., at target 350M FLOP, OFA finds an architecture with 74.93% accuracy while evolutionary search finds a hybrid model that achieves 77.91% accuracy.
- *Hybrid models pareto dominate single models at any target FLOP.* In figure 4(a), hybrid model for 150M FLOP outperforms the OFA search baseline beyond the target 150M FLOP. Similar observation can be made for 250M and 350M FLOPs.
- *Hybrid model achieves near SoTA accuracy with ≈ 3X less FLOPs.* For ex., using a base with 146M FLOPs and 71.68% accuracy, the hybrid model achieves 76.1% accuracy at 80% coverage. To achieve 76.1% accuracy, a single model requires > 450M FLOPs.

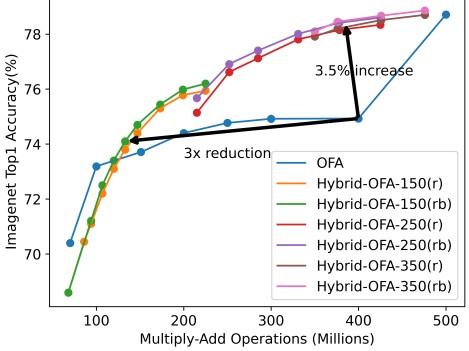

| Base MACs | R / RB | Cov.=90 Acc. (%) | | Cov.=80 Acc. (%) | | Cov.=70 Acc. (%) | |
|---|---|---|---|---|---|---|---|
| | | Base | Hybrid | Base | Hybrid | Base | Hybrid |
| 68M | R | 74.2 | 71.1 | 79.2 | 73.1 | 84.1 | 74.4 |
| 68M | RB | 74.4 | 71.2 | 80.3 | 73.4 | 84.6 | 74.7 |
| 146M | R | 77.1 | 74.2 | 82.1 | 76.1 | 86.5 | 77.1 |
| 146M | RB | 77.6 | 74.4 | 82.9 | 76.4 | 87.3 | 77.4 |
| 225M | R | 79.7 | 76.1 | 84.7 | 77.4 | 88.9 | 78.2 |
| 225M | RB | 79.9 | 76.3 | 84.8 | 77.7 | 89.1 | 78.5 |

Figure 4: Evolutionary search based hybrid OFA-models with hybrid FLOP constraints: 150M, 250M, & 350M. Figure plots intermediate FLOPs achievable by three different hybrid models. Table shows hybrid accuracy and base accuracy at different coverage levels.

- *Abstaining base model achieves significantly better performance than the base at full coverage.* For ex., a base model with $225M$ FLOPs achieves an accuracy of $79.86\%$ when it only covers $90\%$ examples. This performance increases to $88.94\%$ accuracy with $70\%$ coverage.

### 3.3 HYBRID MODELS UNDER OVERWHELMING RESOURCE CONSTRAINTS

In this experiment, we explore hybrid models where base is severely resource constrained. Concretely, we consider ImageNet classification on a tiny MCU (STM32F746 MCU with 320KB SRAM and 1MB Flash), the same seting as MCUNets Lin et al. (2020). Using MCUNet TFLite model (12.79M FLOPs, $51.51\%$ accuracy) as base, we create a hybrid model by adding various OFA global models. The resulting hybrid model performance is given in Table 3, with the energy comparison Appendix. B.4. Below we summarize the benefits of deploying hybrid models:

- *SOTA Accuracy and Pareto Dominance.* Hybrid model with deployable base achieves near SoTA accuracy with $\approx 3X$ fewer FLOPs. In addition, hybrid models consistently dominate stand-alone designs across different target accuracy levels.
- *Micro-controller Implementation.* Successfully deployed base and routing models on a micro-controller with negligible ($\sim 2\%$) slowdown.

Table 3: Hybrid models with MCUNet as base (12.79M FLOP, $51.51\%$ accuracy, deployable on STM32F746 controller with 320KB SRAM & 1MB Flash), operating at various coverage levels.

| Global | Coverage=90 Top1 Accuracy (%) | | Coverage=80 Top1 Accuracy (%) | | Coverage=70 Top1 Accuracy (%) | | Coverage=60 Top1 Accuracy (%) | |
|---|---|---|---|---|---|---|---|---|
| | Base | Hybrid | Base | Hybrid | Base | Hybrid | Base | Hybrid |
| OFA-125M | 55.89 | 56.11 | 60.52 | 60.37 | 65.71 | 64.52 | 71.01 | 67.99 |
| OFA-595M | 55.91 | 57.01 | 60.83 | 62.01 | 65.68 | 66.69 | 71.01 | 70.77 |

### 3.4 HYBRID MODELS UNDER LATENCY METRIC

For simplicity, we used the coverage (i.e., data kept on the base) as a proxy for the communication latency since it is the major contributor to inference cost in a hybrid system. In this section, we explicitly define the communication device and measure the three latency components in this hybrid system: (a) inference latency spent on-device, (b) communication latency for examples sent to cloud, (c) inference latency on the cloud. Table 4 benchmarks the latency ( a+b+c ) of the hybrid approach against various baselines. It shows that the hybrid model operates at nearly half the latency and power consumption compared to the global only solution and provides $8\%$ improvements over the best available model on the device. We provide details and other configurations in the Appendix §B.6.

Table 4: Latency and power consumption: Base device communicates via LoRAWAN (1kbps transmission speed) and operates at 215MHz with 2.5V power supply and active-mode consumption of 100mA per second.

| Base + Global | Method | Params | Top-1 | MACs | Latency | Energy |
|---|---|---|---|---|---|---|
| MCU | Global only | 9.1M | 79.93 | 595M | 1200ms | 300mJ |
| STM32F746 | On-Device | 0.6M | 51.5 | 12.8M | 197ms | 49mJ |
| + | On-Device | 0.74M | 62.6 | 82M | 1075ms | 269mJ |
| GPU Tesla V100 | Hybrid@70Cov | - | 66.69 | 191M | 557ms | 139mJ |
| | Hybrid@60Cov | - | 70.77 | 250M | 677ms | 169mJ |

## 4 CONCLUSION

We proposed a novel hybrid edge-cloud network to handle the majority of the workload of large-scale prediction on edge devices. Currently, large-scale prediction tasks are exclusively handled at the cloud with high-capacity DNNs, and although recent works propose methods for compression of high-capacity models, the resulting models when required to achieve SOTA accuracy are still too large. Our proposed solution is based on leveraging a low-capacity network that can be deployed on an edge device, along with a high-capacity network deployed in the cloud. When the edge device encounters challenging inputs, these inputs are transmitted and processed on the cloud. We proposed a novel end-to-end framework for optimizing network architectures, network models, as well as the routing protocol in a systematic manner. Our proposed method demonstrates substantial decrease in the number of overall floating point operations (FLOPs) on ImageNet dataset compared to a well-designed high-capacity network, while suffering no excess classification loss. Furthermore, when communication latency to the cloud is the dominant issue, we show that across different target accuracy regimes, we realize $4\times$ FLOP gains on the low-capacity model with 70% coverage.

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

# APPENDIX

## A ALGORITHMS

We summarise the methodological proposals as algorithms. The overall method is to begin with training a super-net in the sense of Cai et al. (2020), for which the methods of their paper can be utilised. This produces a set of architectures $\mathscr{A}$, with associated canonical models for each $\alpha \in \mathcal{A}$. The overall procedure then is summarised as Algorithm 1. This uses the two main procedures of architecture search (Algorithm 4) and hybrid training (Algorithm 3) as subroutines, which in turn may be executed in a modular way as discussed at length in the main text.

In addition, we frequently tune a given router $r$ and base and global models to locally trade-off resource usage levels and accuracy (which saves on retraining on each different value of $\varrho$ that one may be interested in. This is realised by finding a value $t$ adjusted to the constraint, and using the routing function $r(x; t) = \mathbb{1}\{r_o(x) \geq r_1(x) + t\}$. Such a $t$ may be found as in Algorithm 2.

---

**Algorithm 1** End-to-end Hybrid Procedure

---

1: **Input:** Training data $B = \{(x^i, y^i)\}_{i=1}^N$, Validation data $V = \{(x^j, y^j)\}_{j=1}^M$, resource constraint $\varrho$.
2: Train supernet using the method of Cai et al. (2020).           *(Architecture Search)*
3: $\mathscr{A} \leftarrow$ resulting set of algorithms.
4: $(\alpha_b, \alpha_g) \leftarrow$ output of Algorithm 4 with $V, \varrho, \mathscr{A}$.
5: Train initial models $b^0 \in \alpha_b, g^0 \in \alpha_g$ using $B$           *(Hybrid Training)*
6: $(r, b, g) \leftarrow$ output of Algorithm 3 instantiated with $B, b^0, g^0$, and with appropriate hyperparameters.
7: **Return:** $(r, b, g)$

---

**Algorithm 2** Tuning Routing Model

---

1: **Input:** Validation data $V = \{(x^j, y^j)\}_{j=1}^M$, target resource level $\varrho$, Hybrid model $(r, b, g)$.
2: $\mathcal{T} \leftarrow \{r_0(x) - r_1(x) : x \in V\}$.
3: $c^* \leftarrow \min c : \mathcal{R}_r + \mathcal{R}(\alpha_b) + (1 - c)\mathcal{R}_g \leq \varrho$.
4: $t^* \leftarrow c^*$th quantile of $\mathcal{T}$.
5: **Return:** $t^*$.

---

**Algorithm 3** Training Hybrid Models

---

1: **Input:** Training data $B = \{(x^i, y^i)\}_{i=1}^N$
2: **Hyper-parameters:** $\lambda_r$, # Epochs $E$
3: **Initialize:** random $r^0$, pre-trained $b^0, g^0$.
4: **for** $e = 1$ **to** $E$ **do**
5:     Randomly Shuffle $B$
6:     $r^e = \arg\min_r \mathcal{L}_{\text{routing}}(r, b^{e-1}, g^{e-1})$
7:     $g^e = \arg\min_g \mathcal{L}_{\text{global}}(r^e, b^{e-1}, g)$
8:     $b^e = \arg\min_b \mathcal{L}_{\text{base}}(r^e, b, g^e)$
9: **Return :** $(r^E, b^E, g^E)$

---

---

**Algorithm 4** Evolutionary Joint Architecture Search

---

1: **Input:** Validation data $B = \{(x^i, y^i)\}_{i=1}^N$, resource constraint $\varphi$, set of architectures $\mathscr{A}$.
2: **Hyper-parameters:** $G, N_{\text{pop}}, N_{\text{par}}$
3: **Initialize:** $\Omega_{\text{pop}} = \{(\alpha_b^i, \alpha_g^i) : \mathcal{R}(\hat{o}, \theta_{\alpha_b^i}, \theta_{\alpha_g^i}) \leq \varphi\}_{i=1}^{N_{\text{pop}}}$ by random sampling
4: **for** $g = 1$ **to** $G$ **do**
5:    $\Omega_{\text{par}} \leftarrow N_{\text{par}}$ highest (oracle) accuracy configurations from $\Omega_{\text{pop}}$
6:    $\Omega_{\text{child}} \leftarrow \emptyset$
7:    **for** $n = 1$ **to** $N_{\text{pop}}$ **do**
8:       Randomly pick $(\alpha_b^i, \alpha_g^i)$ from $\Omega_{\text{par}}$
9:       $(\alpha_b^m, \alpha_g^m) \leftarrow \text{Mutate}(\alpha_b^i, \alpha_g^i)$
10:      Compute the agreement oracle $\hat{o}$ for $\theta_{\alpha_b^m}, \theta_{\alpha_g^m}$.
11:      **if** $\mathcal{R}(\hat{o}, \theta_{\alpha_b^m}, \theta_{\alpha_g^m}) > \varphi$ **then**
12:        GOTO 9.
13:      Add $(\alpha_b^m, \alpha_g^m)$ to $\Omega_{\text{child}}$
14:    $\Omega_{\text{pop}} = \Omega_{\text{par}} \bigcup \Omega_{\text{child}}$
15: **Return :** $\Omega_{\text{pop}}$

---

# B  IMPLEMENTATION DETAILS

## B.1  HYPER-PARAMETER SETTINGS.

We use SGD with momentum as the default optimizer in all our experiments. We initialize our hybrid models from the corresponding pre-trained models and use a learning rate of $1e-4$ for learning base and global models. We use a learning rate of $1e-2$ for learning the routing network. We decay the learning rate using a cosine learning rate scheduler. As recommended in the earlier works, we use a weight decay of $1e-5$. We set the number of epochs to be 50. We use a batch size of 256 in our experiments.

## B.2  MODEL DETAILS

**Entropy Thresholding Baseline.** As per recommendation in the literature (Teerapittayanon et al., 2017; Gangrade et al., 2021) we compute the entropy $H$ of the base prediction probability distribution $b_y(x)$. This baseline allows access to a tunable threshold $t$. Predictions with entropy below this threshold are kept with the base model while the predictions with entropy above this threshold are sent to the cloud model. We use similar tuning as Algorithm 2 to trade-off resource usage.

**Routing Model.** Our routing model uses predictions from the base model and creates a 2-layer neural network from these predictions. We create meta features from these predictions to reduce the complexity of the network, by (a) adding entropy as a feature, (b) and adding correlations between top 10 predictions, resulting in a 101 dimensional input feature vector. The feed-forward network has 256 neurons in the first and 64 neurons in the second layer. The final layer outputs a two dimensional score leading to binary prediction for the routing $r$. Note that the routing network described in this manner contributes to less than 2% compute budget of the base model and hence its compute cost is negligible in comparison to the base and global models.

**MobileNetV3.** We have used the small and large configurations as base and global models in our experiments (see Sec. 3.1 ). We borrowed pre-trained models from publicly available implementation [1]. Table 5 lists the performance and compute characteristics of these borrowed models.

**Once-for-All.** We borrowed the pre-trained OFA models from the official public repository [2]. Table 6 lists the accuracy, number of parameters and FLOPs for these models. We note that these models have been specialized by the authors with fine-tuning to achieve the reported performance.

---

[1] https://github.com/rwightman/pytorch-image-models
[2] https://github.com/mit-han-lab/once-for-all

Table 5: MobileNetV3 Models in our setup.

|                     | Top1 Accuracy | #Params | #MACs  |
|---------------------|---------------|---------|--------|
| MobileNetV3-Small   | 67.613        | 2.54M   | 48.3M  |
| MobileNetV3         | 73.3          | 3.99M   | 143.4M |
| MobileNetV3         | 71.7          | -       | 112M   |
| MobileNetV3         | 70.4          | -       | 91M    |
| MobileNetV3-Large   | 75.721        | 5.48M   | 215.3M |

Table 6: Once-for-All Pre-trained models in our setup.

|                                                          | Top1 Accuracy | #Params | #MACs |
|----------------------------------------------------------|---------------|---------|-------|
| OFA-600 ('flops@595M_top1@80.0_finetune@75')             | 79.9          | 9.1M    | 595M  |
| OFA-482 ('flops@482M_top1@79.6_finetune@75')             | 79.6          | 9.1M    | 482M  |
| OFA-389 ('flops@389M_top1@79.1_finetune@75')             | 79.1          | 8.4M    | 389M  |
| OFA-230 ('LG-G8_lat@24ms_top1@76.4_finetune@25')         | 76.4          | 5.8M    | 230M  |
| OFA-151 ('LG-G8_lat@16ms_top1@74.7_finetune@25')         | 74.6          | 5.8M    | 151M  |
| OFA-101 ('note8_lat@31ms_top1@72.8_finetune@25')         | 72.8          | 4.6M    | 101M  |
| OFA-67 ('note8_lat@22ms_top1@70.4_finetune@25')          | 70.4          | 4.3M    | 67M   |

## B.3 ONCE-FOR-ALL SEARCH EXPERIMENTS.

In our evolutionary search experiments (see Sec. 3.2), we have used the OFA search to create the baseline models that eliminate the effect of fine-tuning available in the pre-trained models from their official repository. We created the baseline by using their optimized search to find models at different FLOPs, namely $\{70M, 100M, 150M, 200M, 250M, 300M, 400M, 500M\}$. We report the performance of these models in the Table 7. Note that as per recommendation, we tune the batch norm statistics of these models to get the correct accuracy.

For our evolutionary search experiments, we used the OFA search space. In OFA codebase, there are two search spaces with MobileNetV3 backbone: (a) with width multiplier 1 and (b) with width multipler 1.2. For our joint architecture search with target FLOPs 150M, we used the smaller backbone with width= 1 as this space allows smaller base models in the < 100M FLOPs region. While we used the backbone with width= 1.2 for our search for target flops 250M and 350M, as these hybrid FLOPs allow larger base models. OFA space allows searching over expansion factor options [3,4,6], kernel sizes [3,5,7], block depths [2,3,4], and resolutions [144, 160, 176, 192, 208, 224]. To perform a mutation, each optimization variable is modified with probability 0.1, where modification entails re-sampling the variable from a uniform distribution over all of the options. The population size is set to 100, and the parent set size is set to 25.

Table 2 shows the characteristics of the base and global models found using this search.

Table 7: Once-for-All models found using the optimized OFA search (used as baseline in Sec. 3.2).

|           | Top1 Accuracy | #MACs |
|-----------|---------------|-------|
| OFA-500   | 78.71         | 500M  |
| OFA-400   | 74.93         | 500M  |
| OFA-300   | 74.92         | 300M  |
| OFA-250   | 74.77         | 250M  |
| OFA-200   | 74.42         | 200M  |
| OFA-150   | 73.71         | 150M  |
| OFA-100   | 73.19         | 100M  |
| OFA-70    | 70.64         | 70M   |

## B.4 MCUNET EXPERIMENTS

We deploy both MCUNet and our base with routing model on the MCU using the TensorFlow Lite for Microcontrollers (TFLM) runtime. Due to lack of operator support for reductions and sorting in TFLM, we replace the relevant operators with supported operations whose compute and memory

complexity upperbounds the un-supported operations. Table 9 compares the performance energy profile of the hybrid model and the baseline when deployed on the micro-controller (STM32F746) with 320KB SRAM & 1MB Flash. It clearly shows that there is a negligible cost of deploying the proposed routing scheme and only results in $< 2\%$ slowdown. Table 8 shows the performance of the hybrid model against the baseline model at various hybrid flops. It can be seen that the hybrid model dominates the baseline model at intermediate FLOPs.

Table 8: Comparing MCUNet models with hybrid models (hybrid accuracy and hybrid flops are shown for hybrid models).

| Target MACs | MCUNet Model | | Hybrid Model | |
|---|---|---|---|---|
| | Top1 (%) | MACs | Top1 (%) | MACs |
| 13M | 51.5 | 12.79M | 51.5 | 12.79M |
| 38M | 57.0 | 38.3M | 56.3 | 36.32M |
| 68M | 60.9 | 67.3M | 62.01 | 67.41M |
| 80M | 62.2 | 81M | 64.52 | 83M |
| 125M | 68.4 | 126M | 71.1 | 127M |

Table 9: Comparing the energy profile for MCUNet and Hybrid model when deployed on a micro-controller.

| Model | Latency | SRAM | Energy |
|---|---|---|---|
| MCUNet | 0.25368s | 156708 bytes | 0.1112 joules |
| Hybrid-MCUNet | 0.25951s | 158036 bytes | 0.1134 joules |

## B.5 Comparison at 70% coverage : Hybrid Model vs Baselines

Fig 2 collates the performance of the hybrid and baseline models from the Experiments section (see Tables 1 and 3, 70% coverage column). Baseline corresponds to the best baseline models at various MACs. Hybrid numbers correspond to the hybrid model where base operates at 70% coverage level. We list the baseline and hybrid performance metrics in Table 10 for completeness.

Table 10: Baseline and Hybrid Metrics used in the Figure 2. Hybrid model is operating at 70% coverage and MACs shown are the Base MACs.

| Model | MACs | | | | |
|---|---|---|---|---|---|
| | 12.8M | 48M | 143M | 215M | 595M |
| Baseline | 51.5% | 67.6% | 73.3% | 75.5% | 79.93% |
| Hybrid | 66.69% | 76.77% | 79.01% | 79.59% | - |

## B.6 Latency Experiments

When we use MCU as the base device, we use the LoRAWAN communication protocol that enables such a low capacity device to operate at a transmission rate of 1kbps. Thus, it takes nearly 1200ms to transfer an image of size 150KB (typical image in the Imagenet dataset). Similarly, for base devices capable of operating with 3G, LTE or Wi-Fi communication devices, we borrow the transmission numbers from Kang et al. (2017). Specifically, to transfer an image with 152KB size, (a) 3G network takes 870ms, (b) LTE takes 180ms, and (c) WiFi takes 95ms. Note that for a micro-controller even a 3G network would not be available, instead a much slower communication device is used. We use the on-device and on-cloud inference latency from the MCUNet and OFA repositories.

For any method, we compute the latency as the inference time taken for an example, i.e., inference time on device + communication time + inference time on cloud. To compute energy usage, we use the active-mode operating characteristics of the base device and voltage supply to compute the power

and multiply it by the amount of time the device spends doing these operations. Table 11 benchmarks the inference latency ( inference cost on device + communication cost + inference cost on the cloud ) of the hybrid approach against various baselines.

Table 11: Deploying a hybrid model vs a standalone model on device. Latency comparison.

| Base + Global | Method | Params | Top-1 | MACs | Latency | Energy |
|---|---|---|---|---|---|---|
| MCU | Global-only | 9.1M | 79.93 | 595M | 1200ms | 300mJ |
| STM32F746 | On-Device | 0.6M | 51.5 | 12.8M | 197ms | 49mJ |
| + LoRAWANN | On-Device | 0.74M | 62.6 | 82M | 1075ms | 269mJ |
| GPU Tesla V100 | Hybrid@70Cov | - | 66.69 | 191M | 557ms | 139mJ |
| | Hybrid@60Cov | - | 70.77 | 250M | 677ms | 169mJ |
| Mobile | Global-only | 9.1M | 79.93 | 595M | 205ms | |
| Samsung Note8 | On-Device | 5.3M | 75.7 | 215M | 65ms | |
| +LTE | Hybrid@70Cov | | 79.59 | 393M | 119ms | |
| GPU Tesla V100 | | | | | | |

## B.7  DIFFERENCE BETWEEN APPEALNET AND OUR HYBRID DESIGN.

Below we highlight main difference between AppealNet (Li et al. (2021)) and our proposal.

- AppealNet formulation does not explicitly model any coverage constraint that enables the base model to operate at a tunable coverage level. In contrast, we explicitly model a coverage penalty.

- Jointly learning the routing without any supervision is a hard problem. Instead, we relax this formulation by introducing the routing oracle that specializes in a routing network for a given base and global pair. With this oracle, the task of learning routing reduces to a binary classification problem with the routing labels obtained from the oracle. This also decouples the routing task from the base and global entanglement.

- In addition, we propose a neural architecture search that finds a pair of base and global architectures that optimise the hybrid accuracy at any given combined resource usage.

- Empirically, AppealNet does not have any evaluations for the Imagenet scale dataset. The closest comparison we can find is with the Tiny-Imagenet dataset (one-tenth of the size of the Imagenet). While we cannot compare the two directly, since we solve a much harder problem than Tiny-Imagenet, we can make the following observations. At $70\%$ coverage level, for AppealNet, the minimum performance difference between the hybrid model and the global model is $\approx 1.2\%$ (see AppealNet, Fig. 5(d)), while our closest to the global in case of the MobileNet baseline is $0.3\%$ (see our paper Table 1, row 3). Note that AppealNet performance will go down on Imagenet in comparison to Tiny-Imagenet due to the hardness of the problem.

