# OpenReview forum: "Hybrid Cloud-Edge Networks for Efficient Inference"
_ICLR.cc/2022/Conference — ICLR 2022 Submitted_

### Official Review · Reviewer_EcwK · 2021-10-26

**Correctness:** 3
**Technical Novelty And Significance:** 3
**Empirical Novelty And Significance:** 3
**Recommendation:** 5
**Confidence:** 3

**Main Review:**

1. The idea of hybird framework to process queries differently based on their difficulty is not a new idea. The basic framework proposed in the submission (big + small model for hard/easy queries and router model for determination) is straight forward and not a very novel idea. Even though the end-to-end training diagram is easy to come up with.
2. What contributes to this framework should be: 1) How to train the router, 2) How to determine the big/small model architectures. Authors address the first question in Section 2.1 and the second in Section  2.2. But in my opinion, explanation in Section 2.1 is a little bit sophisticated: the algorithm should be straight forward but the writting seems to complicate it. I am still confused about the details of how to train the router. Could author formulate the whole process in an Algorithm format?
3. For the determination of model architecture, author proposed to use OFA, which is a seperated work.  But author emphasize in techinical contributions that "propose a NAS method ...". I don't think using OFA can be described as "propose" in the writting. Instead, NAS and the proposed hybird framework is decoupled: NAS does not take advantage of any properties in the hybird framework, vice versa.
4. I may miss some important points in the submission. Please remind me if necessary.

Questions:
1. As author mentions in the related work that this work is similar to dynamic network for they share a similar idea that "different queries should be processed by different network". Though these two kinds of methods differ a lot, especially the proposed method can be deployed much easier, I am still interested in the comparison experiments on dynamic networks. Noted that this is not compulsory.
2. Since the determination of models is handled by another work (OFA), can I regard the submission as the hybird framework as big/small model (given) + router model training. If so, did author try another router model training methods? Different kinds of router models (if applicable) should be compared.

**Summary Of The Paper:**

This paper proposed a hybird framework to process inference efficiently: the framework contains a global network to deal with hard query, a base model for easy query and a router protocol to deliver queries. It designed a proxy supervision algorithm to train the router and proposed to use Neural Architecture Search (NAS) to search for global/base models.

**Summary Of The Review:**

The overall contribution is not significant as the hybird framework is straight forward and the model determination is decoupled and directly from published work.

---

> ### Author Response · Authors · 2021-11-23
> **Addressing specific reviewer comments**
>
>
> We thank the reviewer for their constructive comments. Below we address the reviewer's main concerns.
>
>  - **Novelty of the proposed solution.** \
>    We refer the reviewer to the main rebuttal thread where we discuss our problem and technical novelty.
>
> - **Method section simplification and Algorithmic procedure** \
>  Due to lack of space, we provide the algorithmic procedures in the Appendix (Algorithms 1-4). We also modified the method section to simplify the language.
>
> - **Confusion between OFA and Joint Hybrid Architecture Search** \
> We point out that our architecture search problem is a joint search for the base and global architectures to maximize hybrid performance. To the best of our knowledge, this architecture search problem is unique to our hybrid setup. Our evolutionary search uses the same routing oracle as the one used in training the hybrid model. We rely on the OFA framework since our search is restricted to the architectures available in the OFA space.
>
> - **Comparison between our method and dynamic neural networks** \
> As discussed in our related work (see Adaptive Neural Networks), dynamic neural networks mainly use entropy thresholding as the confidence operator. For a fair comparison, we have added this as a baseline in our experiments (see Fig. 3 and Table 2). We also point out that dynamic neural networks can be used as a base or a global model for additional savings.
>
> - **Our technical contributions and router baselines** \
> We beg to differ from the reviewer on the novelty of the architecture search and refer the reader to the main rebuttal thread where we explicitly state our technical novelty.
> Without the architecture search component, our proposal boils down to learning the routing, base, and global models to maximize hybrid accuracy at a given resource constraint. Note that in this case, we fix the base and the global architectures apriori. In addition, in our architecture search section,  we perform an evolutionary search algorithm to find the base and global model pairs that maximize the hybrid performance at a given resource constraint. We use entropy thresholding as a baseline to compare against the proposed routing network.

---

> > ### Comment · Reviewer_EcwK · 2021-11-30
> > **Thanks for your patient response**
> >
> > I have decided to keep my score for the following reasons:
> >
> > 1. Motivation and algorithm novelty is the most important part (for me) in a research paper. The proposed method show great improvement and practical feasibility on real scene, as shown in "Addressing main reviewer concerns (2/2)". But the core part (OFA+routing) behind the solution is not novel enough.
> >
> > 2. To prove that OFA is mere an option for simplicity and does not harm the contribution, try to do the two things: 1) Try other search space and the proposed method still works. 2)  Find out any inherent property between OFA and proposed method.
> >
> > Best regards.

---

### Official Review · Reviewer_37D1 · 2021-11-02

**Correctness:** 3
**Technical Novelty And Significance:** 3
**Empirical Novelty And Significance:** 2
**Recommendation:** 5
**Confidence:** 4

**Main Review:**


**Strengths**
1. It is interesting to have models with different MACs on cloud and edge to improve the inference accuracy with minimal latency increase.
2. The idea of introducing a routing model to route more challenging inputs to the cloud is quite novel.
3. The paper proposes to provide a better tradeoff between the latency and accuracy, by leaving the more challenging inputs to the clouds with higher latency.

**Weakness**
1. The goal is to improve the inference accuracy by predicting the more challenging inputs on the high latency cloud. It is weird to only focus on the MACs of the models while evaluating the system. How the system affects the latency of edge inference should be shown in the paper.
2. The router model seems only to be updated with soft constraints, which may make the whole system unreliable in practice, since it does not guarantee the average latency (how many inputs are sent to the cloud).
3. Table 7 is the only experiment that indicates the latency of the hybrid inference system proposed, but no detailed experiment setup is given, including the network bandwidth and the computation resources on the edge and cloud.
4. In the technical contribution and experiment section, the author mentioned the 70% communication latency reduction (3 out of 10 examples to the cloud), but it is <20% in the abstraction.
5. It would be better to have a detailed comparison of the edge, cloud, and hybrid inference in terms of latency taking the computation latency of the edge, cloud, and router model into consideration.


**Minor Issues**
1. The MACs and FLOPs are different terms. FLOPs are typically 2x MACs. It would be better to use the same term in the paper.
2. There are several missing related works that combine edge and cloud inference together.
   [1] Huang, Yangsibo, et al. "Instahide: Instance-hiding schemes for private distributed learning." ICML, 2020.
   [2] Liu, Zhijian, et al. "DataMix: Efficient Privacy-Preserving Edge-Cloud Inference." ECCV, 2020.



**Summary Of The Paper:**

The paper proposes to combine the inference on edge and cloud together, taking advantage of the communication-free inference on edge and the high-accuracy model on cloud. It utilizes the OFA to obtain models with different resource requirement at low cost. It achieves higher accuracy on the ImageNet dataset compared to the edge-only inference.

**Summary Of The Review:**

The paper proposes an interesting idea to combine the edge and cloud inference together, but as mentioned above, it is not practical to use MACs as a metric when the main concern of the paper is to maintain the low latency of edge inference while improving the accuracy. The experiments should be redesigned to make the proposed method more convincing.

---

> ### Author Response · Authors · 2021-11-23
> **Addressing specific reviewer comments**
>
> We thank the reviewer for their constructive comments. Below we address the reviewer's concerns.
>
>  - **Latency experiments.** \
>  In the main rebuttal thread, we have shown the relation between coverage and latency. We explicitly define latency and provide detailed experiments. In addition, we have added these experiments in Section.3.4 and with setup details in Appendix B.6.
>
> - **No guarantee on the average latency (how many inputs are sent to the cloud)** \
> We model the coverage constraint ( i.e., how many inputs are sent to the cloud) explicitly in our routing loss function. Our framework allows tuning the threshold $t$ associated with the routing network. In practice, using a validation set, the routing network can be adjusted to achieve a  desired target coverage level.
>
> - **Table 7 details** (table number got changed to $8$ in the revised version)\
> As mentioned in the Section.3.3 and Appendix B.4, in this experiment we create a hybrid model with a severely resource constrained edge device. We borrow the base model ($12.8$M MACs, $51.5\%$Top-1 accuracy) from the MCUNet repository. The latency in the Table only compares the overhead of operating the routing network in addition to the base model. In this experiment, we still use the coverage metric as the proxy for communication. Table 3 shows the hybrid performance at different coverage levels. In this rebuttal, we have included the latency experiment that includes the communication cost for this base and global configuration (see Section 3.4 and Appendix B.6).
>
> - **Mismatch between coverage numbers in abstract and introduction**\
> In our empirical evaluation, we use three different coverage levels $90\%, 80\%, 70\%$ (see Table 1, 3, and Figure 4). In Figure 2, we point out that the accuracy improvements are corresponding to $70\%$ coverage levels. While in the abstract, we mention correct improvements for the $80\%$ coverage level, i.e. $20\%$ abstentions. For consistency between introduction and the abstract, we will update the abstract with the improvements corresponding to $70\%$ coverage level, i.e., $30\%$ abstentions.
>
> - **Related works: Instahide and DataMix**\
> We thank the reviewer for pointing out these works. Although the setup in both the works involve an edge and a cloud device, these works do not aim to maximize hybrid performance at a given resource constraint. Their main contributions in this hybrid setup is to preserve privacy when relying on the cloud inference.

---

> > ### Comment · Reviewer_37D1 · 2021-11-30
> > **Thank you for the detailed responses**
> >
> > I appreciate the detailed response from the authors. The latency table looks pretty good to me, and I would like to increase the score to 5. The authors proposed an interesting idea, but the paper needs to be in a better form.
> > 1. The whole paper spends a lot of space comparing the proposed method with the OFA. It would be better to talk more about the motivation and the ablation of the proposed routing method.
> > 2. The tables and the figures should be redesigned to illustrate contributions more clearly.

---

### Official Review · Reviewer_p5u2 · 2021-11-02

**Correctness:** 3
**Technical Novelty And Significance:** 2
**Empirical Novelty And Significance:** 2
**Recommendation:** 5
**Confidence:** 4

**Main Review:**

Strengths:

The problem they consider is highly significant for practical application of DNNs on edge devices.

Weaknesses:

From the introduction: “The fundamental drawback of cloud ML, however, is the increased latency and energy consumption arising from communication, which can be prohibitive for many applications”, “we propose a best-of-both hybrid solution, which allows for deploying cloud-based AI tasks on edge devices like MCUs, while lowering the average total latency (the sum of communication and computational latency)”, but it is not clear if the proposed method reduces energy consumption. It would be nice to give some comparison of the energy cost of classifying an example on device vs. the energy cost of transmitting a sample to a cloud service and then retrieving the result.


From the introduction: “the best available model which can be deployed on an STM32H743 MCU achieves 62.2% accuracy with 12.8M FLOPs (Lin et al., 2020)”, but Table 4 in (Lin et al., 2020) states that MCUNets achieve 70.7% accuracy on the exact same device. Even their baselines achieve 68% accuracy which is more than 62.2%.

From the introduction: “The base model is compact and designed for devices with low-resource hardware constraints like latency and memory usage”. Latency is not a device constraint, it is an application constraint. I suggest using clock frequency instead of latency.

It is not clear how the Hybrid model was trained and what the baseline is in Fig2. There should be a clear explanation for this figure in the main text..

It is not clear to me what ‘base limited setting’ and ‘global limited setting’ mean in the last sentence of the introduction. A clear explanation is needed.

A parameter “t” is introduced which “allows a routing model to trade-off accuracy and resource usage in order to avoid separately training for each desired level”. How much does it degrade the performance compared with end-to-end training of components for each desired level? An ablation study is needed.

In Section 2, “Architectures and Costs”: “we investigate communication limited settings in §3.3.” Where is this investigated? I cannot find the associated experiment.

The experiments section starts with analyzing results of the experiments and then goes into the description of the experiments, which is a poor writing choice.

“Neurosurgeon: Collaborative Intelligence Between the Cloud and Mobile Edge” tries to tackle the same problem by dividing the inference dynamically among the edge and the cloud. How do you compare your method with it?

“AppealNet: An Efficient and Highly-Accurate Edge/Cloud Collaborative Architecture for DNN Inference” Tries to tackle the same problem with a very similar solution. How is your method different from theirs?

Comments:

I think it makes the last sentence of the abstract more clear if “substantially” comes after the word “accuracy” instead of where it is right now.

I think it is good to give some numbers for on-device computation latency, cloud computation latency, and also communication latency so that the reader develops a sense of them.

Fig1 should change to Fig2 in Empirical validation (last subsection of intro).

I’m curious about the task of learning a router model. Is it a well-defined task? It is interesting to show some validation results in that specific task to see if the router model learns something generalizable and reasonable.

In the “overall formulation”, the constraint is on R(r, b, g), which is the overall cost per inference of models deployed on both the edge device and the cloud. But based on the overall development, the edge device is the only compute limited part and the cloud is an over-provisioned server. I think having a constraint on R(r, b, g) instead of R_r + R(\alpha_b) creates some confusion.


**Summary Of The Paper:**

This paper introduces a hybrid Cloud-Edge design for Deep Neural Network (DNN) inference in which only a small portion of samples (more complicated ones) are sent to cloud for processing. Inference for most of the samples happens on the edge devices itself which reduces the task’s latency.
The paper provides an end-to-end training methodology for system’s components, a Neural Architecture Search (NAS) approach for the design of models deployed on the edge and the cloud. It also does an empirical evaluation of its design.

**Summary Of The Review:**

I think the authors are thinking about a very interesting problem, but the work is at an early stage and not ready for an ICLR publication. The main reason is the lack of comparison with similar work. For more details, see the cons above.

---

> ### Author Response · Authors · 2021-11-23
> **Addressing specific reviewer comments (1/2)**
>
>
> We thank the reviewer for their constructive comments. Below we address the reviewer's main concerns.
>
> - **Communication Limited Settings, Latency and Energy experiments.** \
>   In the main rebuttal thread, we have shown the relation between coverage and latency. We explicitly define latency and provide detailed experiments. In addition, we have added these experiments in Section.3.4  with setup details in Appendix B.6.
>
> - **Comparison with Neurosurgeon**
> 	- Given a fixed high-cost model, Neurosurgeon splits the computation between the device and cloud model by processing the initial part of the network on the device and processing the remaining on the cloud.
> 	- It is a very computationally expensive approach compared to our proposal, as every model incurs communication overhead. In their case, the communication cost is not just the input data (image) but a set of feature maps that will be much larger and have more redundancy. For instance, a typical image would consist of $224 \times 224 \times 3$ floats, while a feature map in their VGG baseline can go up to intermediate stage $224 \times 224 \times 64$.
> 	- In our case, there will be many instances where we do not route the example to the global model, say at a coverage level of $80\%$, our routing scheme on average will only route $20\%$ examples to the global model. In contrast, Neurosurgeon aims to split the computation of a given model between the edge device and cloud, i.e. to achieve accuracy of a global model, the total computation for this high-cost model will be used irrespective of the input example.
> - **Comparison with AppealNet**
> 	- AppealNet formulation does not explicitly model any coverage constraint that enables the base model to operate at a tunable coverage level. In contrast, we explicitly model a coverage penalty.
> 	- Jointly learning the routing without any supervision is a hard problem. Instead, we relax this formulation by introducing the routing oracle that specializes in a routing network for a given base and global pair. With this oracle, the task of learning routing reduces to a binary classification problem with the routing labels obtained from the oracle. This also decouples the routing task from the base and global entanglement.
> 	- In addition, we propose a neural architecture search that finds a pair of base and global architectures that optimise the hybrid accuracy at any given combined resource usage.
> 	- Empirically, AppealNet does not have any evaluations for the Imagenet scale dataset. The closest comparison we can find is with the Tiny-Imagenet dataset (one-tenth of the size of the Imagenet).  While we cannot compare the two directly, since we solve a much harder problem than Tiny-Imagenet, we can make the following observations. At $70\%$ coverage level,  for AppealNet, the minimum performance difference between the hybrid model and the global model is $\approx 1.2\%$ (see AppealNet, Fig.5(d)), while our closest to the global in case of the MobileNet baseline is $0.3\%$ (see our paper Table 1, row 3). Note that  AppealNet performance will go down on Imagenet in comparison to Tiny-Imagenet due to the hardness of the problem.
> - **Incorrect numbers for MCUNet baseline in the introduction.**  \
> 	 We thank the reviewer for pointing out the error in our citation. We borrowed the pre-trained models from the MCUNet github repository (https://github.com/mit-han-lab/tinyml/tree/master/mcunet). We seem to have mixed the controllers STM32F746 (320kB SRAM, 1MB Flash) and STM32H743  (512kB SRAM, 2MB Flash).
> 	- On STM32H743, MCUNet model (467M FLOPs, 3.3M Params) achieves $70.7\%$ Top-1 accuracy.
> 	- On STM32F746, MCUNet model (12.8M FLOPs, 0.6M Params) has 5FPS latency and achieves $51.5\%$ Top-1 accuracy.  We have access to STM32F746 controller and we have used this 5FPS latency model in our experiments. We have clarified this statement in the paper.
> 	- MCUNet authors did not release the deployment implementations, instead we rely on their released tf-lite binaries for execution using the tf-lite library. The 5FPS model is the only one that does not lead to OOM error (see "MCUNet under different latency constraints" on their github ).
> 	- Once again, we re-iterate the point that our hybrid model only improves by using an improved base model. In this case, if a better base model existed than 12.8M FLOPs achieving 51.5\% Top-1, our hybrid performance will only increase.
> - **Latency is not a device constraint, it is an application constraint. Suggests using clock frequency instead of latency in introduction.** \
>  We believe there has been a slight misunderstanding of the way we use the term device constraint. We do not mean a constraint imposed on the device. Instead, we mean a constraint imposed on the model by the device. Our use of the term device constraint is in line with the popular use of that term in the literature (OFA, MCUNet, FBNet, etc.).

---

> > ### Author Response · Authors · 2021-11-23
> > **Addressing specific reviewer comments (2/2)**
> >
> > - **Figure 2 details.** \
> > Figure~2 collates the performance of the hybrid and baseline models from the Experiments section (see Tables 1 and 3, $70\%$ coverage column). Baseline corresponds to the best baseline models at various MACs. Hybrid numbers correspond to the hybrid model where base operates at $70\%$ coverage level. We list the baseline and hybrid performance metrics in Table 10 (see Appendix B.5) for completeness.
> > - **Ablation study for the parameter $t$. How much does it degrade the performance compared with end-to-end training of components for each desired level?** \
> > Due to a lack of time and compute resources, we have not included the end-to-end training of components at each desired level. We point out that training a hybrid model for a coverage level would yield marginally better performance for the specific coverage but would be a computationally expensive exercise.
> > - **Is learning a router well-defined task. Does the router generalizes.** \
> > Learning a routing network is a hard-task. We tackle this problem with a routing oracle that reduces this learning problem into a binary classification task. Router learnt using this procedure generalizes well. For instance, while training a hybrid model with pre-trained MBV3-small and MBV3-large models, on the oracle labels, the router achieves a training accuracy of $\approx 87\%$ and this translates into a validation accuracy of $\approx 84\%$. In contrast, entropy thresholding on the validation dataset achieves $\approx 77\%$ accuracy on the oracle labels.
> >
> > - **Confusion on the including global in the resource constraint in "Overall Formulation" Eq.3** \
> > We point out that Eq.3 shows a generic formulation where the resource constraint depends on $g$. One scenario where this formulation is helpful is in the situation where communication cost is of the order of the global inference cost. In such a case, resources consumed at the global model needs to be accounted in the optimization.

---

### Official Review · Reviewer_Jnva · 2021-11-03

**Correctness:** 3
**Technical Novelty And Significance:** 3
**Empirical Novelty And Significance:** 3
**Recommendation:** 6
**Confidence:** 4

**Main Review:**

As edge devices are severely resource constrained, models (with low capacity) deployed on them don't achieve the same level of accuracy on many prediction tasks as their high capacity cloud counterparts. This work attempts to improve the accuracy of prediction tasks on edge devices by employing a hybrid approach where a low capacity model is deployed on the edge device and a counterpart high capacity model is deployed on the cloud. A query routing model running on the edge device decides which query needs to be handled by the cloud model and thereby achieving a trade-off between classification accuracy and cloud computation/communication costs. All the three models are learnt end-to-end from training data for the desired trade-off.


Positives:
A principled solution to the problem by modelling it as a coupled maximization problem. This involves  a) solving the architecture search problem using evolutionary search and proxies for accuracy of the architecture and b) learning the hybrid model using alternating optimisation.
Principled solution to the routing problem through supervised learning of the routing oracle.
Empirical results that demonstrate that the proposed approach can give competitive results compared to cloud models while reducing the computation required substantially.

Negatives:
Only one dataset is used in the empirical study.
From the graphs, it appears that rbg and rg are very marginally better than r. This means that end-to-end training of all the three models is not really helping much.

The effectiveness of the proposed approach is not only contingent on the predictive accuracy of the routing model (it needs to do a good job of identifying challenging queries (on which the low capacity model is likely to fail) but also on most queries being relatively less challenging (so that the low capacity model can make accurate predictions on them). This implies that predictive tasks that are inherently challenging for low capacity models are unlikely to benefit from the hybrid approach as most queries would need to be routed to the cloud model thereby making it cloud-heavy.


**Summary Of The Paper:**

As edge devices are severely resource constrained, models (with low capacity) deployed on them don't achieve the same level of accuracy on many prediction tasks as their high capacity cloud counterparts. This work attempts to improve the accuracy of prediction tasks on edge devices by employing a hybrid approach where a low capacity model is deployed on the edge device and a counterpart high capacity model is deployed on the cloud. A query routing model running on the edge device decides which query needs to be handled by the cloud model and thereby achieving a trade-off between classification accuracy and cloud computation/communication costs. All the three models are learnt end-to-end from training data for the desired trade-off.


**Summary Of The Review:**

The work is interesting as it addresses an important practical problem and proposes a principled solution to it. Experimental study can be made stronger by considering additional tasks and datasets. Also, practical usefulness of the proposed approach needs to be carefully reasoned as it appears to be critically dependent on most queries being easy for the edge model whereas in practice this might not be the case.

---

> ### Author Response · Authors · 2021-11-23
> **Addressing specific reviewer comments**
>
>
> We thank the reviewer for their constructive comments. Below we address the reviewer's main concerns.
>
>
> - **Predictive tasks that are inherently challenging for low capacity models are unlikely to benefit from the hybrid approach** \
> As discussed in related works, learning with abstention literature shows that the model performance can be increased substantially by abstaining on some examples. In this work, we extend such a framework to route the abstained examples to a high capacity model. Similar routing spirit can be found in dynamic neural networks.
>
>   In addition, our framework provides a tunable threshold $t$ to control the coverage constraint, i.e. how many examples are routed to the cloud. Currently, to achieve near cloud performance, the device has to route all the inputs to the cloud (inference cost becomes too high) or use a baseline that utilizes entropy thresholding (our hybrid scheme outperforms this baseline). In contrast, hybrid design enables trading off coverage for near global performance.
>
>
>  - **rb and rbg are marginally better than r.** \
>  While the plots do not show a significant gap between the three components. A closer look at the hybrid model with MBV3-small and MBV3-large architectures reveals that at $\approx 180$M hybrid MACs,  we see that the 'r' achieves $75.63\%$ accuracy while 'rb' and 'rbg' achieve $75.9\%$ and $76.1\%$ Top-1 accuracy. In this case, the gap between 'r' and 'rbg' is nearly $0.5\%$.  On the Imagenet dataset, this is a non-trivial improvement. We also agree with the reviewer that marginal gains between $r$ and other components need some investigation. We conjecture that the proposed approach is hitting an upper bound in terms of performance.

---

> > ### Comment · Reviewer_Jnva · 2021-11-23
> > **Thanks for your detailed response to the review**
> >
> > "We also agree with the reviewer that marginal gains between  and other components need some investigation."
> >
> > Thanks.  Will look forward to this.

---

> > > ### Author Response · Authors · 2021-11-30
> > > **Additional Details**
> > >
> > >
> > > Some more anecdotal examples where RBG achieves non-trivial improvements over R:
> > >
> > >  - **Accuracy improvements**: In Figure 3, for MobileNetV3, all the RBG models between $70-180$M MACs achieve $\geq 0.5$% improvements in accuracy over R. Similarly, for OFA, most of the RBG models between $100-200$M MACs achieve $\geq 0.7$% improvements over R.
> > >
> > > - **Latency improvements**: In Figure 3, for MobileNetV3, to achieve $75.4$% accuracy, RBG uses $60$% coverage while R utilizes $50$% coverage. It shows that RBG gains $10$% improvements in the latency metric.
> > >
> > > We will include these metrics along with additional details in the next iteration.

---

> > > > ### Comment · Reviewer_Jnva · 2021-11-30
> > > > **rbg vs r**
> > > >
> > > > Improvement in latency is indeed encouraging. You should definitely discuss in the final version in some detail the merits and demerits of relatively more expensive rbg training vs much simpler and cheaper r training.

---

### Author Response · Authors · 2021-11-23
**Addressing main reviewer concerns (1/2)**


We thank the reviewers for their constructive comments. We have updated the paper with appropriate modifications. Below we answer main reviewer concerns and address specific comments in reviewer threads. To summarize our discussion below, we point out that our depicted scenario is novel (not present in other works), is technically sound and new, and does address the cases pointed out by the reviewers.

### Problem Novelty
 - **Our focus is on meeting constraints to deploy models on low-resource edge devices**. This contrasts with prior work including the ones pointed out by reviewers. These works are only concerned with budgeting more resources for complex examples without worrying about the more important device-specific hard constraints (FLOPs, communication latency) imposed by edge devices. We highlight these aspects in the related works while comparing adaptive neural networks.
 - **We propose and study end-to-end system-wide objectives**. Our focus on edge devices necessitates posing a system-wide hybrid objective, and we systematically integrate and optimize all degrees of freedom (architectures, routing \& coverage, base, and global networks). In contrast, prior works optimize these aspects in a decoupled and isolated manner.

### Technical Novelty
While earlier works have tackled similar problems, our technical novelty lies in the following aspects.

 - **Improved training of router.**  Learning a good router is difficult since the optimal routing affects the base and global performance.
	 - *Prior approaches*.  Earlier works either use simplistic post-hoc methods like entropy thresholding or relax the router $r$ in Eq.5. These yield sub-par performance.
	 - *Our approach*. Instead, we transform the routing into a binary classification task by using the routing oracle (see $2.1). It enables us to use standard ML tools for learning good binary functions, thus gaining over approaches that directly try to relax the objective of Eq.5 . Further, our alternating minimisation (see Appendix Algorithm~3) allows training all or a subset of the three components (base, global, and router).
- **Novel Architectural Search Problem.**
	- Our hybrid design raises the novel problem of jointly searching for the base and global architectures to maximise performance under resource constraints.
	- We design an evolutionary search algorithm for this using the routing oracle to provide fast proxies for the fitness of joint architectures.
	- Note that the method is general, and our empirical search is restricted to the OFA architecture space only for simplicity.
	- OFA should be viewed as a search space for different architectures and the search is an integral component within our end-to-end optimization framework. Therefore, it is incorrect to view our contribution as limited to routing (as commented by the reviewer EcwK).

### Latency Experiments.

- **Coverage is a proxy for communication latency, and is well studied in the paper**
	- Communication latency is the major contributor to inference cost in a hybrid system. For instance, on an MCU, the cost of transferring an image to the cloud is $6\times$ the cost of on-device inference.
	- Coverage represents the fraction of points at which no communication is used, and is thus a direct proxy for the savings in communication latency. We have demonstrated advantages in this respect in a variety of settings (Tables 1,2,3; Figure 4).
	- In contrast to the cloud-only solution, our hybrid design gains $3.3\times$ in latency with little drop in accuracy (see Table~1, $70\%$ coverage column).
- **More detailed and explicit modeling.** We augment the rebuttal with the following:
  - We explicitly define communication devices and measure three latency components in the hybrid system
	  A) Inference latency on-device.
	  B) Communication latency for examples sent to the cloud.
	  C) Inference latency on the cloud.
  - Tables below benchmark the latency (A + B + C) of the hybrid approach against baselines.
  - In model IoT type settings - computing on a micro-controller with LoRAWAN communication - our hybrid method
	  - Operates at about **half the latency and power consumption versus a cloud-only** solution.
	  - Gives **$8\%$ improvement in accuracy over** the best available **on-device** model. Note that even $1\%$ gain in accuracy is considered non-trivial on Imagenet dataset (see \cite{Sandler_2018_CVPR_MobilenetV2}).
	  - Detailed results below.
  - In model mobile phone settings with an LTE connection the hybrid model
	  - Operates at **half the latency of a cloud-only solution**.
	  - Achieves near-cloud only performance, that is, $4\%$ improvement in accuracy versus the best available **on-device model**.
	  - Detailed results below.

---

> ### Author Response · Authors · 2021-11-23
> **Addressing main reviewer concerns (2/2)**
>
>
> |   Base+Global  |    Method    | Params | Top-1 |  MACs | Latency | Energy |
> |:--------------:|:------------:|:------:|:-----:|:-----:|:-------:|:------:|
> |       MCU      |  Cloud-Only  |  9.1M  | 79.93 |  595M |  1200ms |  300mJ |
> |    STM32F746   |   On-Device  |  0.6M  |  51.5 | 12.8M |  197ms  |  49mJ  |
> | + LoRAWANN     | On-Device    | 0.74M  | 62.6  | 82M   | 1075ms  | 269mJ  |
> | GPU Tesla V100 | Hybrid@70Cov |        | 66.69 | 191M  | 557ms   | 139mJ  |
> |                | Hybrid@60Cov |        | 70.77 | 250M  | 677ms   | 169mJ  |
> ||||||||
> |     Mobile     |  Cloud-Only  |  9.1M  | 79.93 |  595M |  205ms  |        |
> |  Samsung Note8 |   On-Device  |  5.3M  |  75.7 |  215M |   65ms  |        |
> |      +LTE      | Hybrid@70Cov |        | 79.59 |  393M |  119ms  |        |
> | GPU Tesla V100 |              |        |       |       |         |        |
> ||||||||
>
> ### Missing References.
>
>  1. *Neurosurgeon* --- This approach splits a large neural net between a device and a cloud model, so that computation begins locally, and then continues on the cloud. In particular **every instance is communicated to the cloud**. This means that **every instance suffers the communication overhead**. In complete contrast, **our work reduces the communication overhead by only using it for difficult instances**. Further, neurosurgeon does not consider edge-device constraints like limited memory.
>  2. *AppealNet (AN)* --- This (unpublished) preprint bears strong similarities to the reference \cite{nan2017adaptive}. While the design is broadly similar, crucial design considerations that we study are missing from AN :
>     - AN has no coverage penalty in the training loss.
>     - The AN router training is entangled with the base and global networks, while we decouple its training using the routing oracle.
>     - The AN model is not evaluated on ImageNet, which is a challenging dataset from both the statistical and computational perspective.
>     The differences between AN and our approach are discussed in greater detail below, but the salient points above serve to distinguish our work from it.
>  4. *InstaHide and Datamix* --- **These focus on privacy, rather than efficiency of inference**. This is orthogonal to our investigation.
>
>
> - nan2017adaptive --- Adaptive classification for prediction under a budget
> - Sandler_2018_CVPR_MobilenetV2 --- MobileNetV2: Inverted Residuals and Linear Bottlenecks

---

### Decision · Program_Chairs · 2022-01-20

**Decision:**

Reject

**Comment:**

This paper aims to improve performance on edge devices by utilizing a large-capacity network in the cloud. To this end, the authors suggest using the routing network that decides whether to use the base model (on the edge device) or the global model (on the cloud). They also propose an overall training scheme for learning not only model parameters, but also network architectures. After the discussion period, 3 reviewers are on the negative side, and 1 reviewer is positive. AC thinks that the authors’ response was not enough to convince the negative reviewers. In particular, AC agrees with the negative comments of reviewers on limited novelty, unclear motivation for the proposed method, and unclear presentations. Overall, AC recommends rejection.